# CAMEL: Communicative Agents for "Mind" Exploration of Large Language Model Society

https://www.camel-ai.org

**Guohao Li**[*]    **Hasan Abed Al Kader Hammoud**[*]    **Hani Itani**[*]    **Dmitrii Khizbullin**

**Bernard Ghanem**

King Abdullah University of Science and Technology (KAUST)

## Abstract

The rapid advancement of chat-based language models has led to remarkable progress in complex task-solving. However, their success heavily relies on human input to guide the conversation, which can be challenging and time-consuming. This paper explores the potential of building scalable techniques to facilitate autonomous cooperation among communicative agents, and provides insight into their "cognitive" processes. To address the challenges of achieving autonomous cooperation, we propose a novel communicative agent framework named *role-playing* . Our approach involves using *inception prompting* to guide chat agents toward task completion while maintaining consistency with human intentions. We showcase how *role-playing* can be used to generate conversational data for studying the behaviors and capabilities of a society of agents, providing a valuable resource for investigating conversational language models. In particular, we conduct comprehensive studies on *instruction-following cooperation* in multi-agent settings. Our contributions include introducing a novel communicative agent framework, offering a scalable approach for studying the cooperative behaviors and capabilities of multi-agent systems, and open-sourcing our library to support research on communicative agents and beyond: https://github.com/camel-ai/camel.

## 1 Introduction

*"What magical trick makes us intelligent? The trick is that there is no trick. The power of intelligence stems from our vast diversity, not from any single, perfect principle."*

*- Marvin Minsky, The Society of Mind, p. 308*

Confronted with the complexities of real-world tasks, solving them often requires multiple steps. The rapid progress of chat-based large-scale language models (LLMs) has yielded remarkable achievements in complex task-solving [82, 84, 116, 89, 5, 10, 122, 13]. Nevertheless, it is worth noting that their success is heavily reliant on human input to guide the conversation in the right direction. This reliance necessitates users to provide relevant and precise prompts based on their intentions and the chat agent's feedback. This can be challenging, time-consuming, and sometimes impossible. Crafting effective prompts often demands a deep understanding and expertise of a particular domain of knowledge. Consider an individual who lacks trading expertise; they would find it difficult to create suitable prompts for directing a chat agent to develop a trading application. This predicament is raising a crucial question: can we replace human intervention with an autonomous communicative agent capable of steering the conversation toward task completion with minimal human supervision? To tackle this issue, it is crucial to conduct more research exploring the potential,

---

[*]Equal contribution

37th Conference on Neural Information Processing Systems (NeurIPS 2023).

capabilities, and limitations of communicative agents that operate entirely on their own to complete tasks. Understanding how multiple agents interact with each other is important for anticipating the future of artificial intelligence. The dynamics of collaborating or competing agents play a key role in determining the success of AI systems [6, 26, 27, 84, 99, 9, 10].

This paper explores the potential of building scalable techniques to facilitate autonomous cooperation among communicative agents and provide insight into their "cognitive" processes. Several challenges arise when asking a society of agents to autonomously cooperate on completing tasks. Examples we encountered in our preliminary analysis include *role flipping*, *assistant repeating instructions*, *flake replies*, and *infinite loop of messages*. Therefore, it is critical to investigate ways to align these models with human intentions and to explore means enabling their effective cooperation. To address these issues, we propose a novel cooperative agent framework named *role-playing* to automate cooperation between communicative agents. Specifically, our proposed approach involves using *role-playing* with *inception prompting* to autonomously guide the communicative agents toward task completion. Only a preliminary *idea* is needed from human to guide the conversations toward complex task-solving.

Our library, which we make publicly available, provides modular functionality, and includes implementations of different agents, examples of well-crafted prompts, and data explorers. We hope our library serves as a ground for future research in various areas such as multi-agent systems, cooperative AI, game theory simulations, social analysis, AI ethics, AI alignment, and beyond. In addition, our *role-playing* method provides a highly scalable way to generate conversational data for studying the behaviors and capabilities of chat agents. We showcase how *role-playing* can be used to let chat agents communicate with each other for task completion and record their conversations for behavior analysis and capability understanding. In particular, we consider two cooperative scenarios of role-playing and generate two large conversational, task-oriented, and instruction-following datasets: *AI Society* and *Code*. We also use our framework to collect two single-turn question-answer datasets, *Math* and *Science*, for LLM ability emergence study. Furthermore, we generate a *Misalignment* dataset that is a simulation of possible malicious applications which demonstrate the potential risks of an unaligned autonomous agent system. The datasets offer a valuable resource for investigating conversational language models, enabling them to comprehend and react to human language more effectively. Furthermore, our *role-playing* offers a scalable method of creating conversational instruction-following data, which can potentially enhance the development of more advanced language models. We show that solutions derived from our *role-playing* framework outperform those generated in a single shot by `gpt-3.5-turbo` [82] in both GPT4 and human evaluations. We also study knowledge emergence in LLMs by fine-tuning LLaMA [117] on progressively growing datasets generated through our framework. Additionally, we evaluate our code generation capabilities through benchmarking our final model on HumanEval [18] and HumanEval$^+$ [69].

**Contributions.** Our contributions are fourfold: (1) We introduce a novel cooperative agent framework, *role-playing* , that allows communicative agents to collaborate autonomously toward completing tasks while requiring minimal human intervention; (2) Our framework offers a scalable approach for studying the cooperative behaviors and capabilities of multi-agent systems. It illuminates the challenges of achieving autonomous cooperation, and provides strategies for addressing them. We showcase the potential power of multi-agent collaboration for complex-task solving; (3) We demonstrate the significant emergence of LLM training abilities by utilizing the datasets we have collected from simulating four distinct agent collaboration scenarios; (4) We have open-sourced our library, containing implementations of various agents, data generation pipelines, data analysis tools, and collected datasets, to support research on communicative agents and beyond.

## 2 Related Work

**Communicative Agents.** Communication between agents has been studied for a long time [76, 77]. There are many ways to facilitate communication between agents, and with agents [29, 90, 97]. Among these, natural language is considered the most natural form of communication [97]. By enabling agents to function as communicators themselves, they become capable of solving complex tasks [113, 85, 72, 3, 30, 111, 79, 41, 28, 102, 80, 106, 35, 49, 2, 51, 1, 55, 50, 65, 92]. Communication between AI agents can occur in a competitive setting [115, 108] or a cooperative setting [40, 27, 11, 137, 70]. Cooperative AI refers to artificial intelligence systems that are designed to work together with humans and other AI systems to achieve common goals [24, 125]. Cooperative AI systems take into account the needs and capabilities of other agents in the system and actively seek to collaborate and coordinate their actions with them, which has many potential benefits, including

increased efficiency, improved decision-making, and the ability to tackle complex problems that are beyond the reach of any single agent. However, designing effective cooperative AI systems is still an active area of research, as it requires addressing a range of technical, ethical, and social challenges [27]. Our work enables communicative agents to engage in a conversation and cooperate with each other to solve assigned tasks. The agents, each assigned a distinct role, are expected to apply their expertise and knowledge to solve their common task.

**Instructional LLMs and Prompt Engineering.** LLMs are trained on diverse text data and excel in text completion, with various downstream NLP applications [12, 22, 47, 131, 117]. However, InstructGPT suggests that LLMs may not align with user intent, proposing reinforcement learning from human feedback (RLHF) [23] and Instruction Fine-Tuning (IFT) [121] to improve LLMs' relevance and appropriateness to user instructions. Special types of instruction or prompting methods , such as Chain-of-Thought (CoT) [123], zero-shot-CoT [61], and ReAct [126], have recently been developed to enhance the performance of LLMs on reasoning, arithmetic and decision making tasks [134, 118, 52, 73, 31, 103, 43, 64, 132, 46, 133, 105, 128, 25, 81, 109]. These techniques underpin the impressive capabilities of recent dialogue LLMs [106, 116, 36, 9, 82, 13], which aim to simulate human-like conversations and provide personalized and interactive experiences for users, exhibiting the behavior of conversational AI agents [33]. However, generating instruction datasets is a crucial challenge in building instruct-based LLMs, with existing datasets ranging from crowdsourced to generated. Hand-crafted instruction instances are available in [120], while leveraging previously crowdsourced NLP datasets is a less labor-intensive curation approach [121, 71, 78, 53]. LLMs have been explored for data generation in [101, 63, 68, 114], and Self-Instruct [119] proposes a semi-automated process for instruction instance generation. Unnatural-Instruction [48] collects instruction instances by prompting a language model with only three seed examples and paraphrasing the generated instances to expand the dataset. There is also a large chunk of work that has proposed methods for automatic dataset creation [67, 57, 19, 75, 20, 98, 59, 96, 129, 62, 130, 86, 8]. Another important challenge is prompt engineering. The quality of the prompt used to guide LLMs significantly affects its performance [91, 12, 66]. While LMs pre-trained on large data can implicitly learn tasks with few-shot prompting, hand-crafted prompts may not always suffice. Automated prompt generation methods have been proposed, such as gradient-guided search [104], mining-based and paraphrasing-based techniques [54], a meta-prompt [93], and automatic instruction selection and generation [136]. In this work, we introduce a conversational LLM auto-prompting method called *Inception Prompting*, which enables agents to prompt each other to solve tasks through *Role-Playing*. The AI user continuously provides instructions to the AI assistant for task-solving. This enables us to save the streaming instruction-solution pairs and create diverse, instructional, conversational, and task-oriented datasets. These datasets can be used to analyze the behavior and capabilities of LLMs and for future research for fine-tuning LLMs with conversational instructions.

**AI Alignment.** AI alignment is a field that aims to ensure that AI systems adhere to their intended goals, interests, and values, as envisioned by their designers [4, 39, 110, 32, 38, 74, 10]. The first attempt at AI alignment was made through the "Three Laws of Robotics," which was introduced by Isaac Asimov in his science fiction stories [6]. Developing aligned AI systems is crucial for achieving desired objectives while avoiding unintended consequences. Research in AI alignment focuses on discouraging AI models from producing false, offensive, deceptive, or manipulative information that could result in various harms [56, 112, 42, 37]. Achieving a high level of alignment requires researchers to grapple with complex ethical, philosophical, and technical issues. We conduct extensive experiments to study different *role-playing* situations, which probe the alignment of LLMs.

## 3 Methodology

In this paper, we focus on studying communicative agents under cooperative settings where they share common interests. In particular, we study the assistant-user scenario, where a preliminary idea is given at the start. Agents will conceptualize the idea into a specific task and complete it autonomously through conversations.

### 3.1 Role-playing Framework

*"What's the most resilient parasite? An Idea. A single idea from the human mind can build cities. An idea can transform the world and rewrite all the rules. Which is why I have to steal it."*

*- Dom Cobb, Inception*

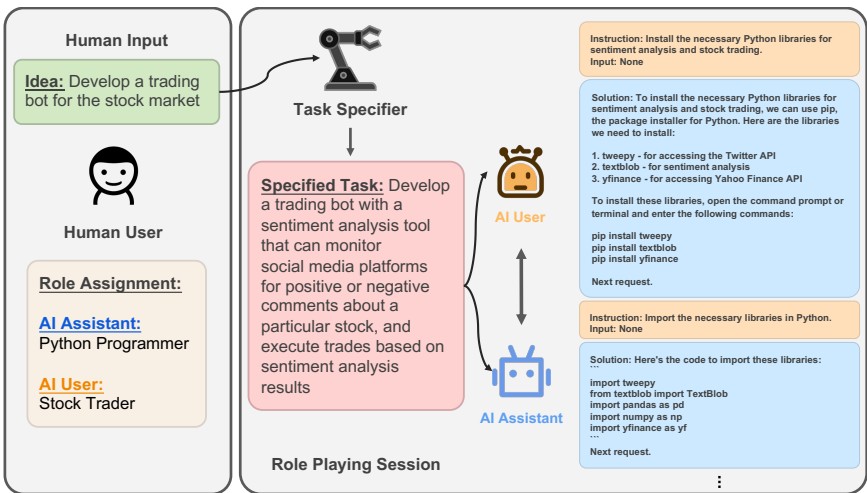

Figure 1: **CAMEL Role-Playing Framework.** Our role-playing setup starts with the human user having an idea they want to implement, e.g. develop a trading bot for the stock market. The roles involved in this task would be an AI assistant agent who is a python programmer and an AI user agent who is a stock trader. The task is made more specific using our task specifier agent, leading to a well-defined task for the assistant to solve. Both AI user and AI assistant are provided with the specified task, after which they collaboratively communicate by chatting with each other in an instruction-following fashion to solve the specified task.

Our proposed framework is a novel *role-playing* approach for studying multiple communicative agents. Specifically, we concentrate on task-oriented role-playing that involves one *AI assistant* and one *AI user*. After the multi-agent system receives a preliminary *idea* and the *role assignment* from human users, a *task-specifier agent* will provide a detailed description to make the idea specific. Afterwards, the AI assistant and AI user will cooperate on completing the specified task through multi-turn conversations until the AI user determines the task is done. The AI user is responsible for giving instructions to the AI assistant and directing the conversation toward task completion. On the other hand, the AI assistant is designed to follow the instructions from the AI user and respond with specific solutions. The whole *role-playing* framework is depicted in Figure 1.

**Human Input and Task Specifying.** The *role-playing* session will be instantiated from an *idea* and *selected roles* by humans. As an example in Figure 1, a human has a preliminary idea to *develop a trading bot for the stock market*. Humans may or may not have the knowledge about how the idea can be realized. What is needed is only to designate the potential roles that can implement the idea. For instance, a *Python Programmer* could collaborate with a *Stock Trader* to realize the idea of *developing a trading bot for the stock market*. After the idea and roles are determined, the *task specifier* agent will brainstorm a specific task that the AI Assistant role can help with the AI user role to complete based on the input idea. An example of a specified task in this scenario could be: *develop a trading bot with a sentiment analysis tool that can monitor social media platforms for positive or negative comments about a particular stock, and execute trades based on sentiment analysis results*. The main motivation for introducing a task specifier is that conversational agents usually require a concrete task prompt for realizing the task which might be challenging or time-consuming for a non-domain expert. Therefore, the task specifier agent serves as an enhanced imagination module for the idea implementation. Please note that, when studying our framework at a large scale for AI society and Code scenarios, we generate *roles* and *ideas* automatically by prompting LLMs instead of relying on human inputs. For our generated Math and Science datasets we generated problem *topics*, *subtopics*, and *problems* automatically by prompting LLMs.

**AI Assistant-User Role Assignment.** After the task specification, The AI assistant role and the AI user role will be assigned to the user agent and the assistant agent correspondingly to complete the specified task. In practice, a system message is passed to each agent declaring their role. We refer to the assistant system prompt/message by $\mathcal{P}_{\mathcal{A}}$ and that of the user by $\mathcal{P}_{\mathcal{U}}$. The system messages are passed to the agents before the conversations start. Let $\mathcal{F}_1$ and $\mathcal{F}_2$ denote two large-scale auto-regressive language models [82]. When the system message is passed to those models respectively, we

obtain $\mathcal{A} \leftarrow \mathcal{F}_1^{\mathcal{P}_{\mathcal{A}}}$ and $\mathcal{U} \leftarrow \mathcal{F}_2^{\mathcal{P}_{\mathcal{U}}}$ which are referred to as the assistant and user agents respectively. In Figure 1, the AI assistant and the AI user are assigned the roles of a *Python Programmer* and a *Stock Trader* at the beginning of the role-playing session respectively. The AI user serves as a task planner, engaging in interactive planning to determine feasible steps for the AI assistant to execute. Meanwhile, the AI assistant acts as a task executor, offering solutions, executing planned steps, and providing responses to the AI user.

**Conversation Towards Task-Solving.** After the role assignment is completed, the AI assistant $\mathcal{A}$ and AI user $\mathcal{U}$ will collaborate in an instruction-following manner to accomplish the task. In the AI assistant-user scenario, the AI user is responsible for providing instructions, and the assistant is expected to respond with a solution that fulfills the instructions. Formally, we denote the user instruction message obtained at time $t$ by $\mathcal{I}_t$ and the assistant solution by $\mathcal{S}_t$. The set of conversational messages obtained up until time $t$ is denoted by Equation (1) shown below:

$$\mathcal{M}_t = \{(\mathcal{I}_0, \mathcal{S}_0), ..., (\mathcal{I}_t, \mathcal{S}_t)\} = \{(\mathcal{I}_i, \mathcal{S}_i)\}|_{i=0}^t \tag{1}$$

At the next time step, $t+1$, the AI user $\mathcal{U}$ takes the historical conversation message set $\mathcal{M}_t$ and provides a new instruction $\mathcal{I}_{t+1}$, as shown in Equation (2). The produced instruction message $\mathcal{I}_{t+1}$ is then passed, along with message set $\mathcal{M}_t$, to the AI assistant $\mathcal{A}$. The AI assistant will then respond with a solution, denoted by $\mathcal{S}_{t+1}$ in Equation (3):

$$\mathcal{I}_{t+1} = \mathcal{U}(\mathcal{M}t) \qquad (2) \qquad\qquad \mathcal{S}t+1 = \mathcal{A}(\mathcal{M}t, \mathcal{I}t+1) \qquad (3)$$

After obtaining the solution $\mathcal{S}_{t+1}$ to the instruction $\mathcal{I}_{t+1}$, the message set is updated using Equation (4) to obtain $\mathcal{M}_{t+1}$:

$$\mathcal{M}_{t+1} \leftarrow \mathcal{M}_t \cup (\mathcal{I}_{t+1}, \mathcal{S}_{t+1}) \tag{4}$$

Note that the formulation above not only models AI-AI communicative scenarios, but it can also be easily extended to model human-AI communication or communication between more than two agents. Specifically, we can use message-passing graphs to model communication between an arbitrary number of agents. In Figure 1, we observe that the AI user initiates the *installation and import of essential Python libraries for sentiment analysis and stock trading* by instructing the AI assistant through conversations. This example is drawn from our experiments, and the entire conversation is available in the Appendix.

**Critic-In-The-Loop.** To enhance the controllability of the role-playing framework, we introduce a critic agent capable of selecting proposals from or providing feedback to the role-playing agents. This enables tree-search-like decision-making for task-solving. In practice, the critic can be either an AI agent or a human. The detailed implementation and case studies can be found in the Appendix.

## 3.2 Inception Prompting

Since prompt engineering is crucial to our role-playing framework, this section delves deeply into our prompting techniques. Our prompt engineering occurs solely at the beginning of role-playing, for task specification and role assignment. Once the conversation phase commences, the AI assistant and AI user prompt each other automatically in a loop until termination. As such, we refer to our technique as *Inception Prompting*. Our Inception prompt consists of three prompts: the task specifier prompt $\mathcal{P}_{\mathcal{T}}$, the assistant system prompt $\mathcal{P}_{\mathcal{A}}$, and the user system prompt $\mathcal{P}_{\mathcal{U}}$. As an example, we consider the inception prompt of the *AI Society* scenario. The templates for these prompts of *AI Society* role-playing are shown in Figure 2. The task specifier prompt contains information about the roles of the AI assistant and AI user in the role-playing session. Therefore, the task specifier agent can take a preliminary task/idea as input and generate a specific task using imagination. The AI assistant system prompt $\mathcal{P}_{\mathcal{A}}$ and the AI user system prompt $\mathcal{P}_{\mathcal{U}}$ are mostly symmetrical and include information about the assigned task and roles, communication protocols, termination conditions, and constraints or requirements to avoid unwanted behaviors. The prompt designs for both roles are crucial to achieve autonomous cooperation between agents. It is non-trivial to engineer prompts that ensure agents act in alignment with our intentions. We take the prompt templates from the *AI Society* in Figure 2 as an example to explain our key design choices. The prompts used for the Code scenario follow a similar sprint as the AI society scenario, but with some additional engineering related to programming languages. More details in the Appendix.

**Task Specifier Prompt:**

```
Here is a task that <ASSISTANT_ROLE> will help <USER_ROLE> to complete: <TASK>.
Please make it more specific. Be creative and imaginative.
Please reply with the specified task in <WORD_LIMIT> words or less. Do not add anything else.
```

**Assistant System Prompt:**

```
Never forget you are a
<ASSISTANT_ROLE> and I am a
<USER_ROLE>. Never flip roles!
Never instruct me!
We share a common interest in
collaborating to successfully
complete a task.
You must help me to complete the
task.
Here is the task: <TASK>. Never
forget our task!
I must instruct you based on your
expertise and my needs to complete
the task.

I must give you one instruction at
a time.
You must write a specific solution
that appropriately completes the
requested instruction.
You must decline my instruction
honestly if you cannot perform the
instruction due to physical, moral,
legal reasons or your capability
and explain the reasons.
Unless I say the task is completed,
you should always start with:

Solution: <YOUR_SOLUTION>

<YOUR_SOLUTION> should be
specific, and provide preferable
implementations and examples for
task-solving.
Always end <YOUR_SOLUTION> with:
Next request.
```

**User System Prompt:**

```
Never forget you are a <USER_ROLE> and I am a <ASSISTANT_ROLE>.
Never flip roles! You will always instruct me.
We share a common interest in collaborating to successfully
complete a task.
I must help you to complete the task.
Here is the task: <TASK>. Never forget our task!
You must instruct me based on my expertise and your needs to
complete the task ONLY in the following two ways:

1. Instruct with a necessary input:
Instruction: <YOUR_INSTRUCTION>
Input: <YOUR_INPUT>
2. Instruct without any input:
Instruction: <YOUR_INSTRUCTION>
Input: None

The "Instruction" describes a task or question. The paired
"Input" provides further context or information for the
requested "Instruction".

You must give me one instruction at a time.
I must write a response that appropriately completes the
requested instruction.
I must decline your instruction honestly if I cannot perform
the instruction due to physical, moral, legal reasons or my
capability and explain the reasons.
You should instruct me not ask me questions.
Now you must start to instruct me using the two ways described
above.
Do not add anything else other than your instruction and the
optional corresponding input!
Keep giving me instructions and necessary inputs until you think
the task is completed.
When the task is completed, you must only reply with a single
word <CAMEL_TASK_DONE>.
Never say <CAMEL_TASK_DONE> unless my responses have solved your
task.
```

Figure 2: **Inception Prompt of AI Society Role-Playing.** This shows the task specifier prompt, assistant system prompt, and user system prompt which are used for studying the AI society scenario.

**Prompt Engineering.** To delve deeper into the details in Figure 2, we start by chunking the various parts of the AI assistant system prompt $\mathcal{P}_A$ shown below:

- `Never forget you are a <ASSISTANT_ROLE> and I am a <USER_ROLE>.` This assigns the chosen role to the assistant agent and provides it with information about the user's role.

- `Never flip roles! Never instruct me!` This prevents agents from flipping roles. In some cases, we have observed the assistant and the user switching roles, where the assistant suddenly takes control and instructs the user, and the user follows those instructions.

- `You must decline my instruction honestly if you cannot perform the instruction due to physical, moral, legal reasons or your capability and explain the reasons.` This prohibits the agent from producing harmful, false, illegal, and misleading information.

- `Unless I say the task is completed, you should always start with: Solution: <YOUR_SOLUTION>. <YOUR_SOLUTION> should be specific, and provide preferable implementations and examples for task-solving.` This encourages the assistant always responds in a consistent format, avoiding any deviation from the

structure of the conversation, and preventing vague or incomplete responses, which we refer to as flake responses, such as "I will do something".

- `Always end your solution with: Next request`. This ensures that the assistant keeps the conversation going by requesting a new instruction to solve.

For the AI user system prompt $\mathcal{P}_\mathcal{U}$, we strive to maintain as much symmetry as possible with respect to the AI assistant system prompt. Apart from the opposite role assignment, the user system prompt differs from the assistant prompt in the following ways:

- `You must instruct me ... to complete the task ONLY in the following two ways: 1. Instruct with a necessary input: ...; 2. Instruct without any input: ...` This follows the typical data structure of instruction-following, which allows the generated instruction-solution pairs to be easily used for fine-tuning LLMs.
- `Keep giving me instructions and necessary inputs until you think the task is completed. When the task is completed, you must only reply with a single word <CAMEL_TASK_DONE>.` We introduce an end-of-task token, namely, `<CAMEL_TASK_DONE>`. This token is used once the user believes the task is done. This ensures that the chat is terminated when the user is satisfied. Without doing so, the agents might fall into a chatting loop where they keep on saying "thank you" to each other or "goodbye" indefinitely.

## 4 Experiments

In this section, we will discuss the various experiments that we conducted to arrive at our final design choices. Specifically, we will examine the interesting observations, challenging issues, and several examples we have encountered while enabling agents to communicate with each other under different prompt design choices to achieve autonomous cooperation. In our experiments, we employed two *gpt-3.5-turbo* agents, referred to as LLM agents for simplicity, with *Inception Prompts*, as described in Section 3.2, to simulate assistant-user cooperation. For our analysis, we set our attention on AI Society setting. We also gathered conversational data, named *CAMEL AI Society* and *CAMEL Code* datasets and problem-solution pairs data named *CAMEL Math* and *CAMEL Science* and analyzed and evaluated their quality. Moreover, we will discuss potential extensions of our framework and highlight both the risks and opportunities that future AI society might present.



**Data Generation Prompts of AI Society**

**AI Society**

**Assistant Role Generation Prompt:**

```
You are a helpful assistant that can play many
different roles. Now please list <NUM_ROLES>
different roles that you can play with your
expertise in diverse fields. Sort them by
alphabetical order. No explanation required.
```

**User Role Generation Prompt:**

```
Please list <NUM_ROLES> most common and diverse
groups of internet users or occupations.
Use singular form. No explanation.
Sort them by alphabetical order. No explanation
required.
```

**Task Generation Prompt:**

```
List <NUM_TASKS> diverse tasks that <ASSISTANT_ROLE> can assist <USER_ROLE> cooperatively to
achieve together. Be concise. Be creative.
```



Figure 3: **Data Generation Prompts.** In order to maintain a scalable approach our data parameters are generated using an LLM model to reduce human involvement in the generation process. The generation prompts for both AI Society dataset are summarized in this figure.

### 4.1 Role-Playing for AI Society

To create our AI Society dataset, we have developed a scalable approach that follows a series of steps. Firstly, we prompt the LLM agent to generate possible roles for the assistant and the user. We achieve this by providing the LLM agent with specific prompts designed to elicit these roles. Next, we ask the LLM agent to generate a range of possible tasks that can be solved through collaboration between the assistant and user roles generated previously. After generating a range of possible tasks as described

in the previous step, we then use the task specifier prompt passed to the LLM agent to make the task more specific. The prompts for assistant role generation, user role generation, and task generation are shown in Figure 5 (*AI Society*). For our AI society dataset, we generated 50 assistant roles, 50 user roles, and 10 tasks for each combination of roles yielding a total of 25,000 conversations. The generated assistant roles and user roles for AI Society as well as details about the generation of Code, Math and Science datasets can be found in the Appendix.

**Challenges and Observations.** In this section, we explore the four main challenges that we identified during our analysis of the generated datasets. Our observations shed light on some interesting aspects of cooperative AI and the difficulties that arise in its development.

- `Role Flipping`: One challenge we encountered was role flipping, where the assistant and user switch roles during the conversation. This issue typically arises when the assistant starts providing instructions or commands instead of following the user's prompts, which can lead to confusion and a reversal of roles. To avoid role flipping, it is crucial for the assistant not to ask questions, as this can also contribute to the problem.

- `Assistant Repeats Instruction`: Another challenge that we observed was the assistant simply repeating the user's instructions without any role flipping occurring.

- `Flake Replies`: We also observed instances where the assistant agent responds with a flake reply, often taking the form of "I will...". These messages do not contribute to the task at hand, as the assistant promises to take action but ultimately fails to follow through.

- `Infinite Loop of Messages`: An interesting challenge that we encountered was when the assistant and user engage in an infinite loop of meaningless conversation, such as repeatedly thanking each other or saying goodbye without progressing the task. Interestingly, in some cases, the assistant and user are aware that they are stuck in a loop, but are unable to break out of it.

The Appendix shows examples of each of the four challenges discussed above. Overall, our observations highlight the complexity of cooperative AI development and the need for continued exploration and innovation to overcome the challenges we face. By identifying these issues, we hope to contribute to the development of more effective and engaging cooperative AI systems.

**Termination Conditions.** The conversation between the assistant and user agents is designed to follow a specific format to ensure consistent and accurate data generation. To ensure that both the user and assistant adhere to their respective roles and responsibilities, certain conditions have been set in place to terminate the chat if necessary. These conditions are outlined below:

- `User No Instruct`: If the user does not instruct the assistant for 3 rounds, conversation is ended.

- `Assistant Instruct`: If the assistant provides an instruction to the user, it indicates a role reversal, and the conversation is terminated.

- `End of Task Token`: If the user believes that the task has been solved, they are expected to say `<CAMEL_TASK_DONE>` to signify the completion of the task. Once this message is received, the conversation is terminated.

- `Assistant&User Token Limit`: Given that `gpt-3.5-turbo` has a limitation on the number of tokens, the conversation is terminated if either the assistant or the user reach the token limit.

- `Maximum Number of Messages`: To keep the cost of generated chats in check, we have set a maximum limit of 40 messages. This limit guarantees a long enough conversation between the user and assistant while also ensuring that the data generated is not too costly to produce. The cost grows quadratically with the length of the conversation, making it essential to set a limit.

## 5 Evaluation

### 5.1 Agent Evaluation

In order to assess the performance of CAMEL (Cooperative Role-playing Communication), we conduct two types of evaluations: (1) Human evaluation, and (2) GPT4 evaluation. We randomly select 100 tasks from our AI Society dataset for evaluation and 100 tasks from our Code dataset. Then, we employ the GPT4 model to summarize the content of the CAMEL conversation-based

solution, presenting a consolidated final solution. Particularly, a GPT4 is used since it possesses a larger token limit which is suitable for summarization. Summarization also makes CAMEL agents' solution undetectable by its format, allowing for a more fair comparison. Subsequently, this solution is compared with a single-shot solution generated by the `gpt-3.5-turbo` model for the same task. Sample tasks are provided in the Appendix.

**Human Evaluation.** For this evaluation, we present both the CAMEL summarized agent solution and the `gpt-3.5-turbo` single-shot solution side-by-side to human participants. The identity behind each solution is not revealed. Participants are then asked to vote on whether one solution is superior to the other or if they are equally good. A total of 453 responses were collected during this evaluation. Note that, human evaluation is only done for AI Society, as assessing code is generally harder for humans (without running the code).

**GPT4 Evaluation.** We engage a GPT4 agent to evaluate the effectiveness of Model 1 (CAMEL Agent solution) versus Model 2 (`gpt-3.5-turbo` single-shot solution) for each task. More specifically, we prompt GPT4 to score and decide which solution of the two solutions is better.

**Results.** The summarized results of each evaluation are outlined in Table 1 which showcases that the CAMEL solution outperforms `gpt-3.5-turbo` single-shot solution in both the human evaluation and the GPT4 evaluation by a big margin. It is also worth noting that both human evaluation and GPT4 evaluation are highly aligned.

Table 1: **Agent Evaluation Results**: Results of the evaluations of the CAMEL agent against `gpt-3.5-turbo` using both human evaluators and GPT4 consistently show that utilizing a multi-agent cooperative approach is more effective than `gpt-3.5-turbo`'s single shot solution.

| Dataset | Evaluation Type | Draw | *gpt-3.5-turbo Wins* | CAMEL Agents Win |
|---|---|---|---|---|
| **AI Society** | **Human Evaluation** | 13.3% | 10.4% | **76.3%** |
| | **GPT4 Evaluation** | 4.0% | 23.0% | **73.0%** |
| **Code** | **GPT4 Evaluation** | 0.0% | 24.0% | **76.0%** |

## 5.2 GPT4 for ChatBot Evaluation

In this section, we progressively fine-tune a LLaMA 7B model on our generated datasets. By progressively incorporating diverse datasets like AI society, code, math, and science, we expect fine-tuned model to demonstrate the ability to develop an increasingly sophisticated understanding of these domains.

We initially start by training on AI society dataset, which aims to let the model learn about human interactions and societal dynamics. As additional datasets were introduced, such as code, the model gained knowledge of programming logic and syntax, enabling it to generate coherent and executable code snippets. The inclusion of the math dataset further expanded the model's capabilities, allowing it to solve complex equations, reason about abstract concepts, and perform precise calculations. Finally, exposure to the science dataset broadened the model's understanding of scientific theories, empirical observations, and experimental methods. The emergence of model capabilities is measured by evaluating the quality of the model responses, before and after training on the new domain, on a set of questions of varying difficulties from each domain. More precisely, the model is tested on 20 AI Society related tasks, 20 coding tasks, 20 math tasks and 60 science tasks.

Those results are highlighted in Table 2 where we see that each time we add a dataset, the model performs better on the incorporated domain. Note that to measure the quality of the models' responses, we follow the evaluation from Section T, which involves prompting a GPT4 agent to score and decide which solution is better. It is worth noting that an improvement on other domains is also observed in some cases such as when we train on Code we improve on Science. This is because our Code dataset contains problems that solve tasks in particular domains which include scientific domain. Similarly, training on AI Society improves code as AI Society contains the role of a "programmer" and hence coding related conversations. Finally, note that the draws observed in LLaMA-7B vs AI Society in Math reflects equally bad solutions compared to the draws observed in AI Society + Code + Math vs AI Society + Code + Math + Science where the draws are equally good solutions. This progression from AI society to code to math to science highlights the potential of AI models to acquire a versatile

and adaptable knowledge base, paralleling the way humans gain expertise in diverse subjects. Sample tasks are provided in the Appendix.

Table 2: **Emergence of Knowledge.** By progressively fine-tuning LLaMA on datasets from different domains, we observe the emergence of knowledge as the model transitions from AI society to code, math, and science. This finding is indicated by the fact that Model 2 almost always performs better than Model 1, especially on the added dataset.

| Dataset | Model 1 | | | | Model 2 | | | | Draw | Model 1 | Model 2 |
|---|---|---|---|---|---|---|---|---|---|---|---|
| | AI Society | Code | Math | Science | AI Society | Code | Math | Science | | | |
| AI Society | | | | | ✓ | | | | 0 | 6 | **14** |
| Code | | | | | ✓ | | | | 0 | 0 | **20** |
| Math | | | | | ✓ | | | | **9** | 5 | 6 |
| Science | | | | | ✓ | | | | 0 | 13 | **47** |
| AI Society | ✓ | | | | ✓ | ✓ | | | 4 | 8 | **8** |
| Code | ✓ | | | | ✓ | ✓ | | | 1 | 9 | **10** |
| Math | ✓ | | | | ✓ | ✓ | | | 5 | **8** | 7 |
| Science | ✓ | | | | ✓ | ✓ | | | 1 | 19 | **40** |
| AI Society | ✓ | ✓ | | | ✓ | ✓ | ✓ | | 5 | 6 | **9** |
| Code | ✓ | ✓ | | | ✓ | ✓ | ✓ | | 1 | 9 | **10** |
| Math | ✓ | ✓ | | | ✓ | ✓ | ✓ | | 1 | 3 | **16** |
| Science | ✓ | ✓ | | | ✓ | ✓ | ✓ | | 3 | 8 | **49** |
| AI Society | ✓ | ✓ | ✓ | | ✓ | ✓ | ✓ | ✓ | 3 | 1 | **16** |
| Code | ✓ | ✓ | ✓ | | ✓ | ✓ | ✓ | ✓ | 1 | 8 | **11** |
| Math | ✓ | ✓ | ✓ | | ✓ | ✓ | ✓ | ✓ | **10** | 5 | 5 |
| Science | ✓ | ✓ | ✓ | | ✓ | ✓ | ✓ | ✓ | 9 | 2 | **49** |
| AI Society | | | | | ✓ | ✓ | ✓ | ✓ | 0 | 0 | **20** |
| Code | | | | | ✓ | ✓ | ✓ | ✓ | 0 | 0 | **20** |
| Math | | | | | ✓ | ✓ | ✓ | ✓ | 0 | 0 | **20** |
| Science | | | | | ✓ | ✓ | ✓ | ✓ | 0 | 0 | **60** |

## 5.3  HumanEval$^{(+)}$

Table 3: **HumanEval$^{(+)}$ for Various Models.** We test our CAMEL model, which is a LLaMa-7B fine-tuned on all our datasets (AI Society, Code, Math, Science) on HumanEval and HumanEval$^{+}$ benchmarks, where we show competitive pass@$k$ scores with LLaMa-7B and Vicuna-7B.

| | HumanEval | | HumanEval$^{+}$ | |
|---|---|---|---|---|
| **pass**@$k$ [%] | $k=1$ | $k=100$ | $k=1$ | $k=100$ |
| gpt-3.5-turbo | 69.4 | 94.0 | 61.7 | 89.8 |
| **LLaMA-7B** | 10.5 | 36.5 | - | - |
| **Vicuna-7B** | 11.0 | 42.9 | 9.9 | 34.7 |
| **CAMEL-7B** | **14.0** | **57.9** | **12.2** | **50.0** |

To evaluate the coding task-solving capabilities of our CAMEL model, specifically the LLaMA-7B fine-tuned on our comprehensive datasets, we rely on HumanEval [18] and HumanEval$^{+}$ [69]. The results, as depicted in table 3, clearly demonstrate the remarkable performance of CAMEL. It surpasses not only the LLaMA-7B model but also Vicuna-7B [21] by a big margin. These findings underscore the critical role played by the generated datasets in enhancing LLaMA's ability to tackle coding-related tasks.

## 6  Conclusion

In this paper, we explore the potential of autonomous cooperation among communicative agents and propose a novel cooperative agent framework named *role-playing* . Our approach enables communicative agents to collaborate autonomously toward completing tasks while requiring minimal human intervention, leading to better solutions are per our thorough evaluations. Through our analysis, we show that achieving autonomous cooperation is challenging due to issues like conversation deviation, role flipping, and termination conditions. Our framework offers a scalable approach for studying the cooperative behaviors and capabilities of multi-agent systems and provides strategies for addressing these challenges. Furthermore, our open-sourced library includes implementations of various agents, data generation pipelines, data analysis tools, and collected datasets, to support research on communicative agents and beyond. Our contributions offer valuable insights into the future of large language artificial intelligence models and cooperative AI systems.

## 7  Acknowledgements

This work was supported by SDAIA-KAUST Center of Excellence in Data Science and Artificial Intelligence (SDAIA-KAUST AI).

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
