# OpenReview forum: "CAMEL: Communicative Agents for "Mind" Exploration of Large Language Model Society"
_NeurIPS.cc/2023/Conference — NeurIPS 2023 poster_

### Official Review · Reviewer_Z1bn · 2023-07-05

**Soundness:** 3 good
**Presentation:** 3 good
**Contribution:** 2 fair
**Rating:** 4
**Confidence:** 4

**Summary:**

This paper introduces CAMEL, a role-playing framework involving two LLMs (an AI user and an AI assistant) communicating with each other to finish a specific task prompted by a human. The two agents are supposed to give instructions and provide answers respectively. On complex tasks (including AI society, code, math, and science), the proposed framework achieves better performance than a GPT-3.5-turbo single-shot baseline evaluated by both human and GPT-4. Further analysis involving fine-tuning a LLaMA model suggest the challenges and emerging behaviors in multi-agent multi-turn conversations.

**Strengths:**

1. This paper introduces an interesting framework role-play that instead of requiring delicate human prompting, two agents are auto-prompted each other (Inception prompting) to solve a task by collaboration in a multi-turn conversation. This may add values to the research community studying complicated prompting methods to solve complex tasks.
2. Evaluation and analysis indicate challenges in multi-agent collaboration such as role flipping and conversation deviation. This can be interesting to future research involving multiple language models.


**Weaknesses:**

1. Baseline and evaluation. The proposed method, although sounds promising and conceptually novel, is not very different from previous methods such as chain-of-thought reasoning, especially React and self-critic, where at each turn, new instructions are prompted and can be considered as "AI user" in this context. The main difference if whether one language model is employed, or two agents in a self-play setup. I agree that there may be some values in using multiple agents as shown in recent works, but using gpt-3.5 with single-shot prompting as the only baseline is not convincing. More importantly, it is not clear what prompts are used for the single-shot baseline and how the prompts are constructed. Furthermore, despite the explanation on why GPT-4 is used to summarize CAMEL before evaluation, this evaluation setting is not convincing because 1. there is no analysis on how much "hallucination" or "error propagation" is generated because of using GPT-4. In other words, the summarization may be biased by GPT-4 sampling results rather than from CAMEL itself. 2. GPT-4 is used as the evaluator (this is less an issue).
2. Many details are missing. For example, where the data and tasks are sampled from to construct the dataset and why they are used for evaluation. I would suggest the authors to specify corresponding analysis instead of "is available in the Appendix" when revising the paper.

**Questions:**

1. Is there an inconsistency in naming? I think the proposed method is named as "role-playing" but is also referred to as "CAMEL".
2. In line 277, how do the assistant and the user know that they are stuck in a loop but are unable to break out?
3. Can you clarify the questions raised above?

**Limitations:**

The authors have addressed the limitations and potential societal impact.

---

> ### Author Rebuttal · Authors · 2023-08-09
>
> Dear Reviewer Z1bn,
>
> Thank you for your careful review and valuable feedback. We appreciate your recognition of the novelty of our proposed framework and your insightful observations.
>
> **Responses to Strengths:** We're pleased to note your agreement on the potential of our inception prompting method and the multi-agent collaboration's challenges.
>
> Weaknesses:
>
> 1. **Baseline and evaluation.**
>
>     **Response:** We respectfully disagree that the proposed method is not very different from previous methods CoT reasoning, React, and self-critic. CAMEL proposes a multi-agent framework that is very different from single agents approaches such as React and self-critic. In this paper, we showcase realizing multi-agent collaboration for task-solving. We agree that React and self-critic can be also used for task-solving. However, the main difference is that our multi-agent framework provides a broader way to create agents for task-solving. For instance, these agents can use different LLM models and tools. They can have different memory or context windows which is not possible to model with single-agent methods. Moreover, our method can be easily extended to other use cases beyond collaboration such as negotiation, competition, and so on. For games with incomplete information, it is necessary to model using a multi-agent framework.
>
>     We understand your concern regarding the choice of baselines and the evaluation method. We agree that a comparison with methods such as CoT, React, and self-critic. In our revision, we aim to add these additional baselines. Here, we present additional comparisons only with Zero-CoT in the table below.
>
>     |                        | **Draw** | **Zero-CoT Wins** | **CAMEL Agent Wins** |
>     |------------------------|----------|-------------------|----------------------|
>     | **GPT4 Evaluation**    | 4%       | 28%               | **68%**                  |
>
>
>     For the single-shot baseline, we used the specified task instruction and default system prompt as the prompts. We acknowledge that this might have been unclear in the paper and will make sure to clarify it in the revision.
>
>     Regarding the use of GPT-4, we understand why the points were brought up. For the GPT-4 summarization, the agent is asked to extract full and complete solutions by looking at the conversation between a user and an assistant with particular specializations. The final solution should be purely based on the conversation. Since our solution generation uses gpt-3.5-turbo which had a 4K token limit (at the time), we had no option but to use GPT-4, since it has a larger token limit (8K), to extract the conversational solutions generated by the CAMEL agents to summarized ones, which aims at making the judge (GPT-4 or human) unable to tell the differences between the solutions by identifying the format of the solution. But we agree that this might not be ideal and will consider alternatives. We will include an analysis of this in the revision.
>
> **2. Many details are missing.**
>
>   **Response:** We apologize for any lack of detail in the main text. We make a further clarification here:
>
>   **Figure 3: Data Generation Prompts** is the prompt for generating roles and tasks for the AI society dataset.
>
>   **D Data Generation Prompts for Code** is the prompts for generating languages, domains, and tasks for the Code dataset.
>
>   **Appendix E Meta Data** is the metadata generated by LLMs for AI Society and Code datasets. 50 assistant roles and 50 user roles are generated for AI Society. 20 programming languages and 50 domains are generated for Code.
>
>   **F Math and Science Datasets Generation Details**: The Math dataset consists of 50K problem-solution pairs. The Science dataset consists of 20K Physics problem-solution pairs, 20K Biology problem-solution pairs, and 20K Chemistry problem-solution pairs.
>
>   This information will be incorporated more explicitly in the paper’s body rather than referring to the appendix.
>
> **Questions:**
>
> 1. **Inconsistency in naming?**
>
>     **Response:** We would like to clarify that CAMEL refers to our overarching framework, which incorporates several techniques including role-playing, inception prompting, and the generation of multi-turn instructional datasets for fine-tuning. Among these, role-playing is a specific technique we propose within CAMEL to facilitate autonomous collaboration between the AI assistant and AI user. We apologize for any confusion caused by our initial wording and will ensure this distinction is made clear in our revised manuscript.
>
> 2. **... Stuck in a loop ...?**
>
>     **Response:** An example is shown in **Appendix G Challenges - Figure 8: Challenges in Role-Playing - Infinite Conversation**:
>
>     > Assistant: \
>     > It seems like you have already received the necessary information to develop the system. Is there anything else you need assistance with? \
>     > \
>     > User: \
>     > No, thank you for your help. ... \
>     > \
>     > ... \
>     > Assistant: \
>     > Goodbye! \
>     > \
>     > User: \
>     > Goodbye! \
>     > ... \
>     > Assistant: \
>     > It seems like we are **stuck in a loop**. Is there anything else I can help? \
>     > \
>     > User: \
>     > No, thank you. Have a great day! \
>     > ...
>
>     The assistant and the user indeed can detect they're stuck in a loop when their responses become repetitive. However, knowing that they are stuck in the loop does not give them access to terminate the program since they are still in their role-play conversations. But this is important for designing the termination conditions. We automatically terminate the loop by checking the frequency of a list of words like termination token *"<CAMEL_TASK_Done>"*, *“thank you”* and *“you’re welcome”*, etc. We will make sure to articulate this aspect more clearly in our revised manuscript.
>
> 3. **Clarify the questions raised above?**
>
>     **Response:** We hope our responses have clarified the issues raised.
>
> Once again, thank you for your constructive feedback.

---

> > ### Comment · Reviewer_Z1bn · 2023-08-19
> >
> > Thanks for the response.
> >
> > Regarding baseline and evaluation.
> > I understand that CAMEL is a different framework from React and self-critic. I was mostly pointing out that those methods can be considered as prompting a single language model, whereas CAMEL uses two language models. Therefore, it is necessary to show some comparison in both methods, results, and pros and cons of each, to illustrate why CAMEL is better. Thanks for the updated results on zero-CoT, but the comparison does not seem to be fair (because there is no prompting for CoT). Methods like React are more comparable. I am thus still not convinced how much CAMEL is better than other methods (conceptually it does have benefits though). Furthermore, I agree that using GPT-4 as an evaluation metric is adapted now, but again using GPT-4 to summarize the results before doing evaluation greatly complicates the evaluation.

---

> > > ### Author Response · Authors · 2023-08-19
> > > **Response to comparison between CAMEL and React**
> > >
> > > Dear Reviewer Z1bn,
> > >
> > > Thank you for your detailed feedback, particularly your observations regarding React and its comparability with CAMEL.
> > >
> > > To address your concerns:
> > >
> > > **Differences between React and CAMEL:** React operates on an "act and environment" paradigm, where it requires specific actions to be taken in a particular environment. In contrast, the primary experiments with CAMEL do not involve such a setup. This foundational difference makes a direct apples-to-apples comparison challenging. While React provides an interface between a language model and an environment to simulate reactions, CAMEL focuses on collaborative interactions between two models without the necessity of an explicit environment.
> > >
> > > **Possible Integrations:** Indeed, CAMEL could conceptually be comprised of two React agents, signifying that the two methodologies can be seen as orthogonal rather than as direct competitors. CAMEL's approach can be interpreted as a high-level collaborative mechanism that could potentially use React-like structures as its constituents.
> > >
> > > **Evaluation and Comparison:** We acknowledge the need to provide clearer comparisons between CAMEL and other methodologies, like React, to elucidate the unique benefits and possible drawbacks of our approach. However, we are not sure how to set up a direct apples-to-apples comparison with React. If you have some specific suggestions in this regard, we will be happy to work towards providing a more thorough analysis.
> > >
> > > Your feedback underscores the importance of making these distinctions clear and motivates us to refine our paper to better articulate these points. We genuinely appreciate your insights and will strive to address them comprehensively in our revisions.
> > >
> > > Warm regards,
> > >
> > > NeurIPS 2023 Conference Submission3376 Authors

---

### Official Review · Reviewer_6oPa · 2023-07-08

**Soundness:** 4 excellent
**Presentation:** 3 good
**Contribution:** 3 good
**Rating:** 8
**Confidence:** 3

**Summary:**

# Summary
## Motivation
Completing tasks by human-in-the-loop is time-consuming. An alternative is to let autonomous agents cooperate to solve tasks.
## Approach
This paper propose to let two LLM agents *role-play* a user and an assistant to solve tasks.
Their data collection approach follows the following steps:
1. For each dataset, they generate a number of user roles, a number of assistant roles and a number of tasks of each combination.
2. For each user, assistant, task combination, they generate the conversation by prompting both roles to iteratively generating instruction and solutions, and doing critic-in-the-loop to improve generations at each step.

They fine-tuned 7B LLaMa models on different combination of datasets. The comparison between models are judged by GPT-4, whose accuracy is validated by small sample human evaluation.
## Results
First, CAMEL (the proposed role-play method between two LLMs) works better than gpt-3.5 as a single model.
Second, in most cases each generated dataset brings improvement to finetuning.
Third, on third-party code datasets HumanEval+, they show that the finetuned small models works better Vicuna.
## Contribution
1. This paper proposes a framework and prompt methods for solving tasks with multiple LLMs with different roles.
2. This paper generates conversation datasets in solving these tasks, which will be helpful for future research.
3. This paper fine-tuned smaller model and showcased the performance on an out-of-domain code generation dataset.

**Strengths:**

# Originality
Using a team of LLMs or generally NNs to solve a task is an interesting problem, but understudied in practical applications. This paper is innovative in that the generated tasks are in general domain, but also related to the agent roles engaged in the tasks.

# Quality and Clarity
As LLMs become accessible, the methods in this can be almost replicated by putting the prompts into the playground. This paper is clearly written, although not self-contained due to the shortage of space, e.g. the description of critic-in-the-loop model is in the appendix, though very important.

# Significance
The results in this paper are significant, except for the unclear parts in experiments (see questions). Due to the unavailability of good metrics when comparing models for general tasks, this paper uses GPT-4 for evaluation. This practice is widely used and has various issues raised by recent papers. Validating the results by doing small sample human evaluation is a reasonable choice.

**Weaknesses:**

1. The "large scale language model society" can be misleading. In the whole paper, the conversations are between two agents with different roles. Readers might expect more agents to participate in decision-making.
2. It is unclear why this approach is different from hierarchical decision-making with the user as the high-level planner, and assistant as the low-level executor. The planner gives high-level instructions, while the low-level executor generates solutions. It is true that there are not many papers on LLMs for hierarchical decision making, however, there is a recent one [1]. This is not essentially a reason to reject this paper, but the authors should make the connection clearer.

[1] Hierarchical Prompting Assists Large Language Model on Web Navigation

**Questions:**

1. Section 5.1, Fig. 4. are the CAMEL agents here GPT-4 based or the fine-tuned model mentioned in 5.2? If it is GPT-4, would the comparison with get-3.5-turbo be unfair?
2. Section 5.3, on which datasets is the CAMEL-7B trained on? All datasets or just code?

**Limitations:**

Yes

---

> ### Author Rebuttal · Authors · 2023-08-09
>
> Dear Reviewer 6oPa,
>
> Thank you for your thoughtful review and positive feedback on our paper. We're pleased to hear the originality and significance of our work is recognized. Below, we address your questions and concerns:
>
>  **Strengths:**
>
> > 1. **Originality** - Using a team of LLMs or generally NNs to solve a task is an interesting problem, but understudied in practical applications. This paper is innovative in that the generated tasks are in the general domain, but also related to the agent roles engaged in the tasks. \
> > 2.  **Quality and Clarity** - As LLMs become accessible, the methods in this can be almost replicated by putting the prompts into the playground. This paper is clearly written, although not self-contained due to the shortage of space, e.g. the description of the critic-in-the-loop model is in the appendix, though very important. \
> > 3.  **Significance** - The results in this paper are significant, except for the unclear parts of the experiments (see questions). Due to the unavailability of good metrics when comparing models for general tasks, this paper uses GPT-4 for evaluation. This practice is widely used and has various issues raised by recent papers. Validating the results by doing a small sample human evaluation is a reasonable choice.
>
>     **Response to Strengths:** We appreciate your acknowledgment of the originality, quality, clarity, and significance of our work. We recognize that the description of the critic-in-the-loop model was limited due to space constraints, and we will strive to provide more clarity within the main body of the paper.
>
>  **Weaknesses:**
>
> > 1. The "large-scale language model society" can be misleading. In the whole paper, the conversations are between two agents with different roles. Readers might expect more agents to participate in decision-making.
>
>     **Response:** We apologize if the term was misleading. Our focus was to demonstrate how a smaller group of agents with well-defined roles could collaboratively solve a task. These LLM agents play different roles that human society has. In the AI society dataset, we collect 50 * 50 pairs of agents with different society roles. In the appendix (P Critic-In-The-Loop), we also show a small society with three agents: a user agent as a Postdoc, an assistant agent as a Ph.D. student, and a critic agent as a Professor. We understand that readers might expect a larger society of agents, and we will work on clarifying our approach.
>
> > 2. It is unclear why this approach is different from hierarchical decision-making with the user as the high-level planner, and the assistant as the low-level executor. The planner gives high-level instructions, while the low-level executor generates solutions. It is true that there are not many papers on LLMs for hierarchical decision-making, however, there is a recent one [1]. This is not essentially a reason to reject this paper, but the authors should make the connection clearer. \
> > [1] Hierarchical Prompting Assists Large Language Model on Web Navigation
>
>     **Response:** We agree with your comparison of our approach to hierarchical decision-making. Thanks for pointing out the missing reference [1]. This paper is indeed relevant but it came out one week after the NeurIPS submission deadline. So we were not able to cover it. The suggested paper [1] proposes a novel hierarchical prompting Actor-Summarizer-Hierarchical (ASH) for web navigation. The action is generated based on a summarized observation by a summarizer instead of the raw observation. The hierarchical modularized design reduces the heavy reasoning burden and improves performance significantly on Webshop tasks [2]. We agree there are some similarities in terms of reducing the difficulty of reasoning complex tasks in both methods. Our intention was to highlight how specific role-playing with multiple agents can lead to a more cooperative solution. We will strengthen this connection by discussing the similarities and differences with the referenced paper on hierarchical prompting and explaining our unique contribution to the field. We also found a more recent (25 Jul 2023) follow-up work WebArena [3] on using autonomous agents for web navigation to be impressive.
>
>     [2] Shunyu Yao, Howard Chen, John Yang, and Karthik Narasimhan. 2022a. Webshop: Towards scalable real-world web interaction with grounded language agents. arXiv preprint arXiv:2207.01206.
>
>     [3] Zhou, S., Xu, F.F., Zhu, H., Zhou, X., Lo, R., Sridhar, A., Cheng, X., Bisk, Y., Fried, D., Alon, U. and Neubig, G., 2023. WebArena: A Realistic Web Environment for Building Autonomous Agents. arXiv preprint arXiv:2307.13854.
>
>
> **Questions:**
>
> > 1. Section 5.1, Fig. 4 are the CAMEL agents here GPT-4 based or the fine-tuned model mentioned in 5.2? If it is GPT-4, would the comparison with get-3.5-turbo be unfair?
>
>     **Response:** We apologize for the confusion. The CAMEL agents used in our experiment were based on gpt-3.5-turbo for role-playing (See line 288), not GPT-4. GPT-4 is used as a judge for the evaluation. We also use GPT-4, since it has a larger token limit, to extract the conversational solutions generated by the CAMEL agents to summarized ones, which is aiming at making the judge (GPT-4 or human) not able to tell the differences between the solutions by the formats. We will further clarify this aspect in our revision.
>
> > 2. Section 5.3, on which datasets are the CAMEL-7B trained? All datasets or just code?
>
>     **Response:** CAMEL-7B was trained on all datasets (AI Society + Code + Math + Science), not just code, which is similar to how Vicuna was trained on diverse datasets that do not include only code. We will include clearer information about this in the revised paper.
>
> Once again, thank you for your insightful comments. They will be of great assistance as we refine our work in the final version.

---

> > ### Comment · Reviewer_6oPa · 2023-08-16
> > **Reply to authors**
> >
> > Thanks authors for their detailed rebuttal. I think the response answers most questions that I had. I have also read the comments by other reviewers and still lean towards accepting this paper.
> >
> > Is it possible to change the "large-scale language model society" in the title? It is still confusing as whether the language models are large-scale or the society is large-scale. A candidate is "large language model collaboration".

---

> > > ### Author Response · Authors · 2023-08-19
> > >
> > > Dear Reviewer 6oPa,
> > >
> > > Thank you for your thoughtful feedback and for considering our detailed rebuttal. We're pleased to know that our responses addressed most of your concerns.
> > >
> > > In regard to your suggestion about the title, we understand your perspective on the ambiguity it presents. We agree that clarity in the title is crucial, and "large language model collaboration" does offer a more direct understanding of the content of the paper. We will ensure that the title is revised to better capture the essence of our work without causing any confusion.
> > >
> > > Once again, we appreciate your constructive feedback and your inclination toward accepting our paper. Your insights have been invaluable in guiding our revisions, and we are grateful for your continued support.
> > >
> > > Warm regards,
> > >
> > > NeurIPS 2023 Conference Submission3376 Authors

---

### Official Review · Reviewer_wYFm · 2023-07-08

**Soundness:** 3 good
**Presentation:** 4 excellent
**Contribution:** 3 good
**Rating:** 7
**Confidence:** 4

**Summary:**

This paper attempts to address a dilemma in leveraging large-language models (LLMs) for solving complex tasks in a collaborative setting: the question of oft-needed human intervention in the equation. More specifically, the authors have come up with an intuitive and novel cooperative agent framework called role-playing that supports effective task completions by collaboration dialogues between agents (LLMs) without extensive human interventions except at inception. Their framework also offers a scalable way to investigate and refine the collaborative capacities of multi- agent systems and they provide detailed analysis and resolution strategies for challenges that come up in such a to- and-fro LLM-instruction scenario. They evaluate their agents exhaustively with state-of-the-art LLMs like GPT 4 as well as with human intelligence. Moreover, they demonstrate task-specific emergence capabilities of smaller-sized LLMs like Llama using their generated user-instruction datasets for various scenarios and domains.

Contributions:
Datasets/Libraries
1. Their publicly available library provides modular functionality and includes implementations of different agents, examples of well-crafted prompts, and data explorers.
2. Two large conversational, task-oriented, and instruction-following datasets: AI Society and Code.
3. Math and Science dataset (QA) and Misalignment dataset (contains simulations of potential risks of such an
uncontrolled autonomous system)
4. These datasets will help investigate other larger language models, allowing such LLMs to communicate more
effectively with human agents.

**Strengths:**

The proposed method (inception prompting) is intuitive, novel, and well-motivated for a novel collaborative task-solving using LLM-agents.

The paper is mostly easy to follow. For instance, Fig. 2 is pretty detailed when it comes to roles and task assignments.

The supplementary material, appendices, and the libraries provided can be crucial for future works in this direction.

The methods section is scientifically sound with effective strategies being discussed for resolving the unaligned idea flows from a role perspective.

The proposed framework can be used to evaluate collaborative problem-solving in crucial domains like classroom learning and education.

**Weaknesses:**

1. Role/task alignment: For the specific task of building an app for stock trading via analyzing sentiments of certain stocks, it seems that the role assigned to the AI user (stock trader) and the message it generates seems counterintuitive since you wouldn’t generally expect stock traders to know environment variables being the first step towards solving this specific task. Instead, a role (like say tech lead/ tech supervisor) seems more fit for the AI user in that example.

2. Fig. 2 confusion: Fig.2 is a little confusing regarding which agent receives the task-specifier prompt (i.e., at the inception level). If I am reading correctly, the multi-agent scenario seems to include two LLMs, but it is not clear who received the starting ‘task-specifier’ prompt immediately. It’ll perhaps improve the readability if the authors can provide some clarity regarding this either in sec. 3.2 or Fig 2

3. Long-distance memory issues: in section 4.1, the authors provide a set of termination conditions that brute- force the collaborative mutual dialogues between the agents. Is it not entirely clear what exactly the authors mean when they claim 40 to be the max-limit of messages to ensure enough length in the conversation history. Was it chosen purely from a pricing perspective? I think the paper can benefit from having some more analysis from a cost/compute perspective. Also, did the authors come across any cases where the agents tended to effectively forget the previous thread of instruction flow (thus leading to sub-optimized task results) because of long-range memory issues that tend to affect such LLM-based agents?

**Questions:**

(Also suggestions)
1. Sec 3.2 makes note of cases of role-switching being observed between the user and the assistant. Are those cases of LLM hallucination anyway correlated with the framing of the prompt? Does it affect certain tasks more than others or is it observed usually at specific points in the sequence of collaborative dialogues?

2. The authors precisely point out challenges such as instruction repetition, flake replies, and infinite loops of messages with examples in the appendices. However, one way of improving the analysis of the paper would be to have a plot/diagram with more details about the distribution of such cases (frequency, places where they tend to occur in the sequence of dialogues, etc). Also, what methods were used to stop/terminate such cases of looping/role-flipping, was it human intervention or did the authors use flags similar to the task termination conditions?

3. On section 5.2, which explains the methods for evaluating ‘emergence’ capacities in smaller LLMs like Llama, what was the motivation behind the sequence of various domains (math, science etc) being a specific way and not any other? Moreover, Table 1 can also benefit from some clarification about what exactly Model 1 and Model 2 are. Although the authors have pointed to sec 5.1 for the evaluation in sec 5.2, it is unclear which one is Llama variant. Also, there might be a case for further experiments with other relatively smaller-sized instruction-based LLMs like T0 [3], FLAN [2] or InstructGPT [3]

**Limitations:**

The authors have explicitly mentioned the limitations and have provided reasonable strategies to avoid any harmful (pricing/social or otherwise) consequences of their work. However, the paper might benefit from a cost- compute analysis in generating the collaborative datasets using LLMs, perhaps as a paragraph in the main paper.

---

> ### Author Rebuttal · Authors · 2023-08-09
>
> Dear Reviewer wYFm:
>
> **Response to Strengths:**
> We appreciate your positive comments on the novelty, motivation, and potential benefit for future work.
>
> **Weaknesses:**
>
> 1. **Response:** We acknowledge the reviewer's observation about the role of the AI user (stock trader). The decision to use the 'stock trader' role was based on the assumption that the AI user stock trader has domain knowledge about stock trading and some basic knowledge of Python. So that the stock trader can collaborate with a Python programmer to achieve the task with its domain knowledge. We see the merit of having the suggested role of 'tech lead/tech supervisor'. Inspired by your suggestion, we propose a more realistic and aligned workflow:
>
>     Stage 1: A stock trader (AI User) collaborates with a tech lead (AI Assistant) to figure out a detailed implementation plan for developing a trading bot for the stock market.
>
>     Stage 2: A tech Lead (AI User) then collaborates with a Python programmer (AI Assistant) on implementing the obtained plan.
>
>     (See the PDF in the global response for the details)
>
>     In this two-stage approach, we see the implementation is more systematic and well-rounded, which includes the designs of Data Storage, Risk Management, Security, and even Testing and Deployment. What would be even more interesting is to introduce an automatic role-generation agent that can generate roles for a given task. We will add the above discussion in the final manuscript.
>
>
> 2. **Response:** We will clarify this in section 3.2 and enhance Fig. 2 to better illustrate the inception prompts. As mentioned in 3.1 Role-playing Framework - Human Input and Task Specifying. The 'task-specifier' prompt is the system prompt for the task specifier agent which receives a preliminary idea and roles from Human input and produces a specified task. The task specifier agent acts as a brainstorming module to help with producing a specified task for the AI User and AI Assistant agents.
>
>
> 3. **Response:** Our decision to limit the number of messages to 40 was mainly cost-related. Even if we provide a set of termination conditions, we still want to put a safeguard to the max-limit of the message. It is because after the task is completed the agents will provide short outputs like "thank you" and "welcome". If no safeguard is set and termination fails, the conversation will only end until it exceeds the token limit, which may end up with thousands of API calls and hundreds of USD dollars cost.
>
>     As for forgetting the previous thread of instruction flow, it is mainly related to agents asking for information that would not be able to perform since the lack of embodiment or physical information such as date, emails, files, location, etc. For instance, an AI user agent asks an AI assistant agent to book a meeting schedule in its calendar. However, the AI assistant agent does not ask for access to the AI user agent's calendar. Then the AI assistant agent will ask for the AI user agent's calendar access. However, we did not provide calendar API accesses to the AI user which cause the forgetting of the thread of instruction flow. This could be solved by providing API access to the embodiment or physical information. In the appendix (O Embodied Agent), we show an example that provides image generation API to an embodied agent which enables it to generate images.
>
> **Questions:**
>
> 1. **Response:** We indeed noticed that role-switching can sometimes be task-dependent. As mentioned above, it is mainly related to agents asking for information that would not be able to perform since the lack of embodiment or physical information such as date, emails, files, location, etc. (See the response to Weaknesses 3)
>
> 2. **Response:** A dataset analysis is provided in the appendix (Dataset Analysis) including some plots for the distribution of these cases. Please see Figures 9 and 10 for the analysis of the distribution of conversation termination reasons and Figure 11 for the flake message distribution. Yes, the checking is similar to the task termination conditions. Role flipping is checked when the assistant instructs the user with keywords like "Instruction" and looping is checked frequency of a list of words like “thank you” and “you’re welcome” etc.
>
> 3. **Response:** The sequence of domains was initially ordered by the time orders of the collection which is relatively arbitrary. We will clarify the details in Table 1. Model 1 and Model 2 are both LLaMA 7B-based models which are trained on different datasets. For instance, LLaMA-7B is the vanilla model, AI Society and AI Society + Code + Math + Science are models trained on the AI Society dataset and a combination of datasets (AI Society + Code + Math + Science) respectively. Moreover, we appreciate your suggestion of comparing with other LLMs like T0 [1], FLAN [2], and InstructGPT [3].
> Below we provide results that show the emergence of knowledge of AI Society-related concepts of a FlanT5 Model. Fine-tuning a FlanT5 on our AI Society data improves the performance of the model on Society related tasks. Additionally, we compare FlanT5 fine-tuned on AI Society data with a LLaMA-7B fine-tuned on the same data and find that they achieve very similar scores with FlanT5 performing slightly better.
>
>     | **Dataset**     | **Model 1**                | **Model 2**                     | **Draw** | **Model 1 Wins** | **Model 2 Wins** |
>     |---|---|---|---|---|---|
>     | **AI Society**  | FlanT5                     | FlanT5 (+AI Society)            | 1       | 0                | **19**               |
>     | **AI Society**  | FlanT5 (+AI Society)       | LLaMA-7B (+AI Society)          | 2       | **10**               | 8                |
>
> **Limitations: Cost-compute analysis ...**
>
> **Response:** We'll add a paragraph addressing this in the revision. Generating all four datasets (AI Society, Code, Math, and Science) cost around 10,000 USD.
>
> Once again, sincerely thank you for your constructive feedback!

---

> > ### Comment · Reviewer_wYFm · 2023-08-17
> >
> > Thank you for the detailed response.  I still recommend acceptance for this paper, but I would like to point out that the change to the described workflow could entail a significant rewrite so I would encourage you to make the revisions as surgically as possible.  As my recommendation is based on the submitted paper, too significant an overhaul to the framing would invalidate reviews based on the original submission.

---

> > > ### Author Response · Authors · 2023-08-19
> > >
> > > Dear Reviewer wYFm,
> > >
> > > Thank you for your constructive feedback and your recommendation for acceptance. We deeply appreciate your recognition of our work.
> > >
> > > We understand and value your concern regarding the extent of revisions. Our intent is to address the comments you and other reviewers have raised without changing the fundamental framing of our paper. We assure you that the core content and essence of the paper will remain intact, and any alterations will be done with precision and caution.
> > >
> > > Your point about the balance between refining the paper and maintaining its original framing is well-taken. We will add a brief description of the workflow and ensure that our revisions are "surgical", targeting specific areas for improvement while retaining the primary structure and content that formed the basis of the original reviews.
> > >
> > > Once again, we appreciate your valuable insights and your recommendation. We are committed to ensuring that our paper maintains its integrity while addressing the concerns raised.
> > >
> > > Warm regards,
> > >
> > > NeurIPS 2023 Conference Submission3376 Authors

---

### Author Rebuttal · Authors · 2023-08-09

To all esteemed reviewers of our NeurIPS 2023 submission,

We express our sincere gratitude for your thorough and insightful reviews of our manuscript. The detailed feedback from each of you provides us with clear guidance on how to refine and improve our work.

**Reviewer wYFm:**

Thank you for recognizing the novelty and intuitiveness of the proposed inception prompting method. Your positive feedback on the clarity of the paper and the potential of supplementary materials for future works is greatly appreciated. We acknowledge the concerns raised about the role/task alignment and the confusion surrounding Fig. 2. Your keen observations about long-distance memory issues with LLM-based agents are especially valuable and will be addressed in our revisions. We attach a **.pdf** file of a two-staged example that addresses the role/task alignment weakness based on your recommendation. The first stage is between a stock trader and a tech lead, and the second stage is between a tech lead and a Python programmer.

> Stage 1: Tech Lead (AI Assistant) v.s. Stock Trader (AI User) \
> Task: Figure out an implementation plan for developing a trading bot for the stock market.


> Stage 2: Python programmer (AI Assistant) v.s. Tech Lead (AI User) \
> Task: Develop a trading bot for the stock market. {Plan obtained from Stage 1}.


We also added an example to show the emergence of knowledge of AI Society-related concepts of a FlanT5 Model. Fine-tuning a FlanT5 on our AI Society data improves the performance of the model on Society related tasks. Additionally, we compare FlanT5 fine-tuned on AI Society data with a LLaMA-7B fine-tuned on the same data and find that they achieve very similar scores with FlanT5 performing slightly better.

| **Dataset**     | **Model 1**                | **Model 2**                     | **Draw** | **Model 1 Wins** | **Model 2 Wins** |
|---|---|---|---|---|---|
| **AI Society**  | FlanT5                     | FlanT5 (+AI Society)            | 1       | 0                | **19**               |
| **AI Society**  | FlanT5 (+AI Society)       | LLaMA-7B (+AI Society)          | 2       | **10**               | 8                |

**Reviewer 6oPa:**

Your feedback regarding the originality of our approach and its potential significance in practical applications is very encouraging. We appreciate your recognition of the clarity of our paper and the potential replicability of the methods. We are attentive to your concerns about the term "large scale language model society" and the need for clarity in distinguishing our approach from hierarchical decision-making. We agree that the mentioned reference [1] and our proposed method have some similarities in terms of reducing the difficulty of reasoning complex tasks. Their connection and discussion will be added to our revisions.

[1] Hierarchical Prompting Assists Large Language Model on Web Navigation

**Reviewer Z1bn:**

We are grateful for your acknowledgment of the innovative nature of our framework and its potential value to the research community. Your observations on the challenges of multi-agent collaboration and the potential issues in our evaluation methods are well-taken. We will certainly delve deeper into the points raised about baselines, evaluation, and the missing details in our paper.

We understand your concern regarding the choice of baselines and the evaluation method. We agree that a comparison with methods such as CoT, React, and self-critic. In our revision, we aim to add these additional baselines. Here, we present additional comparisons only with Zero-CoT in the table below:

|                        | **Draw** | **Zero-CoT Wins** | **CAMEL Agent Wins** |
|------------------------|----------|-------------------|----------------------|
| **GPT4 Evaluation**    | 4%       | 28%               |  **68%**                  |

In conclusion, we are truly thankful for the time and effort each reviewer dedicated to evaluating our submission. Your feedback is instrumental in guiding us toward improving the quality and impact of our research. We look forward to addressing all raised concerns in our revised manuscript.

Warm regards,

NeurIPS 2023 Conference Submission3376 Authors

---

### Decision · Program_Chairs · 2023-09-21

**Decision:**

Accept (poster)

**Comment:**

This work proposes to accomplish tasks autonomously with LLMs but asking them to role play necessary entities for planning and execution. Ideally, one imagines that a human would interact via dialogue with a system to guide the process, but here, the LLM "acts" the part. The results are successful in the domains evaluated and the approach outperforms appropriate baselines.  The key things I would re-raise from the reviews are: 1. Clarifying early that _two_ agents are interacting and we have limited evidence for the strength of a "society" and 2. Evaluation here is based on GPT-4, which while it aligns with humans in the study, is not a guarantee of success in true execution.